

# Impacts of land use and invasive species on native avifauna of Mo'orea, French Polynesia

Vanessa M. ZoBell[1,2] and Brett J. Furnas[2,3]

[1] Department of Wildlife, Fish, and Conservation Biology, University of California, Davis, CA, United States of America
[2] Department of Environmental Science, Policy, and Management, University of California, Berkeley, CA, United States of America
[3] Wildlife Investigations Laboratory, California Department of Fish and Wildlife, Rancho Cordova, CA, United States of America

## ABSTRACT

Oceanic islands are among the most endemically biodiverse ecosystems in the world. They have been adversely impacted by human expansion, which affects regional biodiversity by altering the natural habitats of vulnerable, indigenous species. Birds represent a valuable indicator species of environmental change due to their ability to adapt quickly. Investigating the relationship between environmental change, abundance, and behaviors of birds can help us better anticipate potential impacts to island ecosystems. In addition, we can understand the population trends and restricted ranges of native avifauna, identify the regions needing protection, and assess habitat vulnerability linked to anthropogenic activities. In Mo'orea, French Polynesia, we studied nine passerine bird species using automated acoustic recording devices placed in agricultural, forested, and mixed habitats. Based on call counts per unit time and occupancy modeling, we found evidence that three non-native species preferred agricultural areas and low-canopy cover over dense forested areas. Furthermore, native bird detectability and possibly abundance was significantly lower than non-native birds. Using hierarchical cluster analysis to support inferences regarding behavioral differences, we found that native bird calling activity was negatively associated with non-native bird calling activity. Altogether, these results suggest native bird populations are at risk in all of the habitats studied, but forests serve as a potential refuge.

## INTRODUCTION

Biodiversity is impacted negatively by human-caused alterations of natural habitats through industrialization, agriculture, logging, and commercial and residential development (*Florens et al., 2012*; *McKinney, 2002*; *Repetto, 1988*). The reduction in floristic and structural diversity caused by agriculture can result in decreased habitat for many animals that rely on natural ecosystems (*Stoate et al., 2009*). Agriculture practices specifically impact birds by altering and eliminating their habitat, nesting availability, and food sources (*Vickery et al., 2001*). For example, the use of fertilizer is known to alter bird occurrence

Corresponding author
Vanessa M. ZoBell,
vmzobell@ucdavis.edu,
vzobell@berkeley.edu

by changing the amount and composition of soil-dwelling invertebrates for consumption (*Scullion & Ramshaw, 1987*; *Tucker, 1992*; *Vickery et al., 2001*). Livestock grazing can also alter biodiversity by removing plants needed for nesting and shelter (*McLaughlin & Mineau, 1995*). Non-native species may reduce biodiversity by encroaching and thriving on new, human-altered habitats such as agricultural areas, and outcompeting their native counterparts for resources (*Mack et al., 2000*; *Gurevitch & Padilla, 2004*).

Oceanic islands are among the most endemically biodiverse in the world because of their isolation. It is their endemism that makes them vulnerable to anthropogenic disturbances (*Paulay, 1994*). French Polynesian oceanic islands host many native and endemic species from molluscs *(Partula mooreana)* and plants *(Nesoluma nadeaudii)* to avifauna *(Ptilinopus purpuratus)* (*Clarke, Murray & Johnson, 1984*; *Meyer & Butaud, 2009*; *Wray, 2013*). On French Polynesian islands, the need for human resources has led to more agriculture areas in recent years (*Kennett, Anderson & Winterhalder, 2006*). When humans colonized French Polynesia in the early Holocene years, they introduced new animals and plants, and altered the land so it could support their population's needs (*Ferdon, 1981*; *Whistler, 1991*). Agricultural intensity increased after the arrival of Europeans who introduced pesticides and other agrochemicals (*Bovis, 1980*; *Sakagawa, 1993*).

Birds play a critical role in conservation, acting as "indicator" species as they adapt quickly to environmental change, demonstrating how other species may change in the future (*Briggs et al., 2013*). By dispersing seeds, they keep the forests rich with plant growth and link habitats that would otherwise remain unconnected (*Galindo-González, Guevara & Sosa, 2000*; *Didham et al., 2005*). Bird-related seed dispersal plays a major role in the genetic exchange in plant populations as well. Should they disappear, birds would not perform ecosystem services, causing detriment to the habitats they live in (*Galindo-González, Guevara & Sosa, 2000*; *Didham et al., 2005*; *Briggs et al., 2013*). Therefore, surveying birds to monitor changes in their distribution and population will help researchers identify impact on the larger ecosystem (*Briggs et al., 2013*).

Despite agricultural conversion of native forests on the island of Mo'orea in French Polynesia, little research has been conducted on the distribution of native and non-native avian fauna on this island. Though the Opunohu Valley on Mo'orea was identified as an Important Bird Area (IBM) by Birdlife International in 2006, a threat score, condition score, and action score have not yet been assessed for the valley (*BirdLife International, 2017*). Therefore, the objective of the present study was to survey nine passerine bird species across forested and agricultural habitats within the Opunohu Valley using automated acoustic recording units. Two of the nine passerine birds, the grey-green fruit dove and the Mo'orean kingfisher, are native (*Wray, 2013*), and Mo'orean kingfisher is endemic to Mo'orea (*Wray, 2013*). Automated acoustic recording units are a valuable technology to help survey a wide variety of avian species concurrently (*Haselmayer & Quinn, 2000*; *Hobson et al., 2002*; *Rempel et al., 2005*; *Brandes, 2008*; *Furnas & Callas, 2015*).

Based on the survey, we evaluate differences in activity and occupancy among native and non-native species, attempt to assess their vulnerabilities to agriculture and loss of forest habitat, and also make recommendations for expanding avian monitoring throughout Mo'orea.

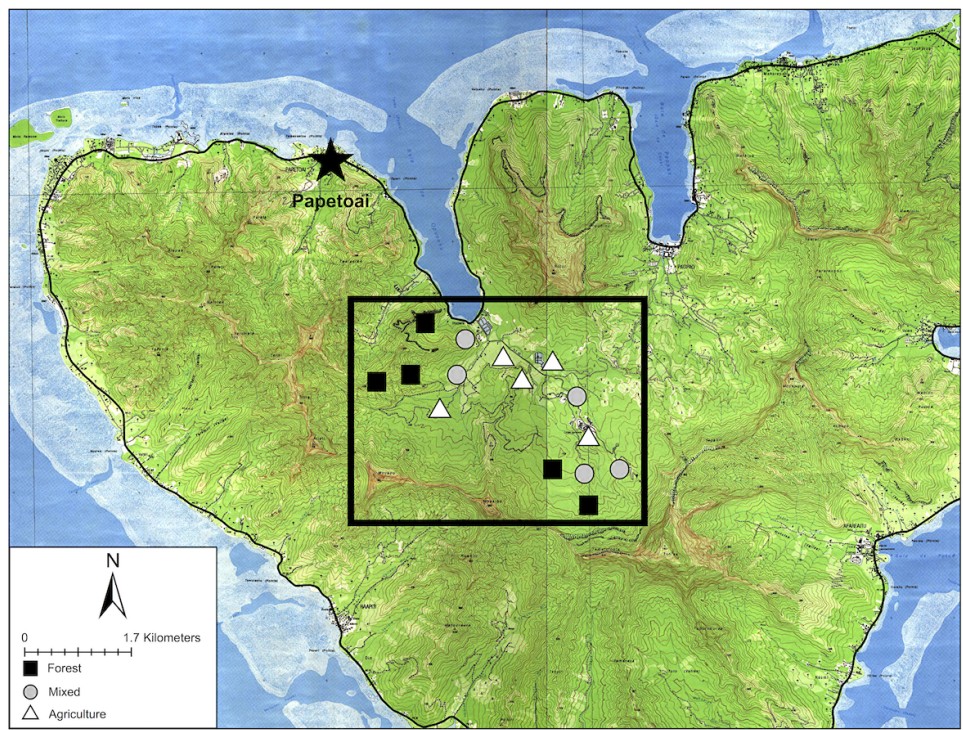

**Figure 1** **Map of Mo'orea with sampling sites.** Sampling sites in the Opunohu Valley of Mo'orea. Black squares indicate forest sites. Gray circles are mixed-covered sites. White triangles are agricultural sites.

## METHODS

### Study area

We conducted this research on the island of Moorea, a high volcanic island, 134 km$^2$ in size, in the Society Islands of French Polynesia (Figs. 1 and 2), where human settlement occurred heavily along the coast of the island as well as in flat river valleys. However, much of the island's rugged interior is unsettled and covered in dense forest filled with Tahitian chestnut *(Inocarput fagifer)*, hibiscus *(Hibiscus rosa-sinensis),* miconia (*Miconia calvescens*), and candlenut (*Aleurites moluccana*). Our bird surveys were limited to the Opunohu Valley, a fertile river valley of diverse land use, from intensive agriculture to protected forests, both interspersed among mixed-used areas. Much of the Opunohu Valley floor currently provides livestock grazing whereas the remnant lowland valley forests are dominated by dense vegetation, some of which is non-native (*Lepofsky, Kitch & Lertzman, 1996*). Farming is intermixed within the Opunohu Valley in multiple areas.

### Study species

The native birds we targeted for acoustic surveys are grey-green fruit dove (*Ptilinopus purpuratus*) and Mo'orean kingfisher (*Todiramphus veneratus*). The introduced, resident avifauna species we surveyed are red jungle fowl (*Gallus gallus*), zebra dove (*Geopelia striata*), red-vented bulbul (*Pycnonotus cafer*), common myna (*Acridotheres tristis*), silvereye (*Zosterops lateralis*), common waxbill (*Estrilda astrild*), and red-browed firetail

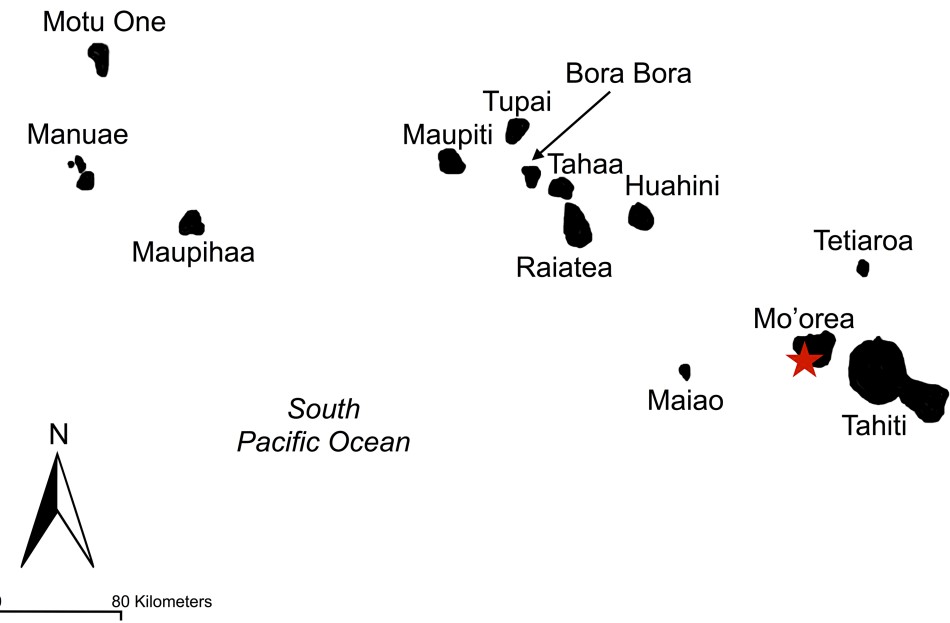

**Figure 2** **Map of Society Islands.** Society islands including Tahiti, Mo'orea, Tetiaroa, Maiao, Huahini, Raiatea, Tahaa, Bora Bora, Tupai, Maupiti, Maupihaa, Manuae, and Motu One. The island sampled, Mo'orea, is marked by a red star.

(*Neochmia temporalis*). Together, these nine species represent all of the terrestrial passerine avifauna on Mo'orea, excluding chestnut-breasted mannikin (*Lonchura castaneothorax*) (*Wray, 2013*). We were not able to distinguish Chestnut-breasted mannikin (*Lonchura castaneothorax*), a terrestrial resident passerine, from spectrogram displays, or find evidence of it being in the agricultural or forested areas, so it was not included in this study. Additionally, the Tahiti swiftlet (*Aerodramus leucophaeus)* has not been seen on Mo'orea since 1973 (*Marshall Cavendish Corporation, 2001*). We did not verify the presence of this species during monitoring, so it was also excluded in this study. We did not include seabirds or shorebirds in our study because they were rarely encountered at the sites we sampled in the Opunohu Valley.

## Selection of field sites

To assess the impact of human land use on the bird community, we first stratified the study area by land-use types (forested, agricultural, or mixed) based on visual inspection of satellite imagery (*Google Earth, 2016*). We overlaid a grid consisting of 267 m wide quadrats and assigned each to one of the three strata. Second, we randomly selected five quadrats within each of the three land use types (15 sites total, Fig. 1). We constrained randomization so that each study site was located >250 m apart to ensure double counting did not occur (*Bibby et al., 2000*). We examined each site in person before sampling to ensure that the habitat type was identified correctly from the satellite imagery.

Ideally, we established a survey site at the center of a selected quadrat. If this point was not accessible, a random distance and bearing was chosen to offset the location. We recorded canopy cover by using five descriptive canopy cover types: open, open/moderate,

moderate, moderate/covered, and covered. Canopy cover was defined by visually assessing the percent of canopy covered by vegetation within an approximately 10 meter by 10 meter box. 0% coverage defined open canopy, 25% coverage defined open/moderate canopy, 50% coverage defined moderate canopy, 75% coverage defined moderate/covered canopy, and 100% coverage defined covered canopy.

## Acoustic sampling

We deployed an automated acoustic recording unit (Olympus DM-620s (Olympus Corporation, Center Valley, PA) or Songmeter4 (Wildlife Acoustics Inc., Concorde, MA, USA)) to record bird vocalizations at each survey site. We rotated use of the two models of recording devices so that an equal number of samples from each model were made in each habitat stratum. This helped us mitigate any issues pertaining to an expected difference in effective sampling area of the two models. The detection range of the acoustic recorders was not measured in our study. However, after attempting visual surveys, we discovered that acoustic recorders in all forested areas and many mixed areas were able to receive calls from birds that were visually obstructed by extremely dense vegetation. For less dense forested habitats in California, *Furnas & Callas (2015)* determined an effective acoustic range of approximately 50 m for the Olympus recorders. It is likely that the detection range of the Wildlife Acoustics recorder is greater than that of the Olympus.

Each automated recorder was placed on the ground within 5 m of the selected point in the quadrat where reception of song was likely to be highest. Adjacent sites were not sampled on the same day to further reduce the chance of recording the same individual from two different sites. In forested sites, we placed the recording device in the clearest area within the 5 m radius so acoustic reception would not be altered by understory vegetation. Following the protocol of *Furnas & Callas (2015)*, we programmed the units to record for five minutes three times each morning commencing at 30 min before local sunrise, sunrise, and 30 min after sunrise. Surveys were repeated on three consecutive mornings totaling nine recordings summing to 45 min at each site. We chose this method to ensure that all species of birds were recorded, taking into account that species may prefer different calling hours. The recordings from our 15 survey sites summed to 675 min from 45 site-morning combinations. The acoustic recorders were set to record at a sample rate of 24 kHz and a reception level minimum of 16.0 dB and a maximum of 122 dB. The bandwidth of the recording devices ranged from 20 Hz to 49 kHz.

## Call validation and interpretation of survey recordings

To ensure accurate interpretation of our survey recordings, we first completed a pilot study to record reference bird calls for each species while visually confirming the species identities of the vocalizing birds. Multiple reference examples were made for each species to ensure variety in call types were identified. We made spectrograms of the validated calls and used them to calibrate species identifications in spectrograms of the recordings from the field.

We extracted the acoustic data from our recording devices via a micro SD card and uploaded them to a computer. All acoustic recordings were converted into an adapted

.wav file format for analysis. Analyses were conducted using the Triton processing software program (*Wiggins & Hildebrand, 2007*), based in MATLAB (MathWorks Inc., Natick, MA), to calculate and display standard spectrograms, to perform audio playbacks, and to log call detections.

One trained acoustic analyst interpreted all of the recordings to minimize any bias that might occur with multiple analysts. The analyst excluded any detections in which she was unsure of the species identification. Less than 20 vocalizations were excluded. We excluded two out of the 45 site-mornings because of intense anthropogenic noise covering bird calls in the spectrograms.

The analyst visually scanned spectrograms by examining a 10 s window with time and frequency resolutions bins of 5 s and 100 Hz respectively (1000 point FFT, 75% overlap). Because of the differences in call structure, defining a single vocalization differed for each species as described below.

Red-vented bulbul calls are most commonly two or three toned, and range from frequencies of 1,000 to 3,000 Hz. Red-vented bulbuls are less commonly found making single tone calls (Supplemental Information). Every every two-tone and three-tone call was considered unique (recorded as a new call) (Supplemental Information). A unique call could be from the same individual, or a new individual. Common waxbill darts are categorized by their high frequency and short duration. The darts range in frequency from 3,000 to 8,000 Hz. The source level is low, and is sometimes, but not often, covered by the high source level of the silvereye call (Supplemental Information). Every dart was considered unique. Zebra dove, like grey-green fruit dove, show a call that contains a train of coos. The coos are faster and of shorter duration than the grey-green fruit dove. Zebra dove coos are also at a higher frequency than grey-green fruit dove, at approximately 1,000 to 1,500 Hz (Supplemental Information). Zebra dove calls are considered unique every time a signature large coo occurs at the beginning of the "coo train." The red jungle fowl call is a combination of sweeps and screeches, varying in frequency, and most commonly found in between 1,000 and 4,000 Hz (Supplemental Information). Every screech was considered a new call. The grey-green fruit dove "coo-train" is low frequency, and varied in duration at approximately 400 Hz (Supplemental Information). The large "coo" at the beginning of the grey-green fruit dove "coo train" determined the beginning a new call. The Mo'orean kingfisher call is categorized as a shuddering "klew" that can vary greatly in duration. The signature shudder of this call was was the largest identification factor. The frequency range of the Mo'orean kingfisher call was very large, at times ranging from 10 to 4,000 Hz in one call (Supplemental Information). The Mo'orean kingfisher call was considered unique if there was a two-second gap between the shuddering "klews." The common myna call was very complex, and was comprised of different trills, screeches, downsweeps, and tonal calls. The common myna, like the Mo'orean kingfisher, has a large range of frequency-calling abilities, and can range from 1,000 to 8,000 Hz (Supplemental Information). The common myna call was considered new if there was a one second gap between the calls. The red-browed firetail call is noted as a high frequency upsweep trill. The upsweep is consistently most intense at a frequency of 6,000 to 8,000 Hz

(Supplemental Information). Each upsweep made was a new call. The downsweep of the silvereye is most commonly found at a frequency of 3,000 to 5,000 Hz (Supplemental Information) and its song is made up of an entanglement of downsweeps at this frequency. Every silvereye downsweep was considered its own call, including all of the downsweeps making up the song. Because of the wide variety of frequency, duration, and call structure in the different species' calls, disentanglement of calls was not an issue in the spectrogram analyses.

## Statistical analyses

We used nested ANOVA to evaluate differences in call count per unit time in different habitats for each species. To this we tallied the number of calls heard per 5-minute survey by habitat and species. We tested if there were any significant differences in mean counts among the habitats by species.

We used hierarchical cluster analysis to further elucidate patterns of co-detection of bird species (*Sharma, 1996*; *McCune, Grace & Urban, 2002*). We limited this analysis to the subset of five sites at which all nine avian species were detected, a constraint which permitted inferences about temporal partitioning of vocal behavior among native versus non-native species. We used logistic regression to confirm temporal difference in species detections.

## Occupancy modeling

Occupancy modeling allowed us to differentiate our data on the species' calling frequencies into separate processes governing the proportion of sites occupied by a species versus the detection probability of a species at an occupied site (*MacKenzie et al., 2006*). We used a multi-species modeling approach (*Tingley et al., 2012*; *Iknayan et al., 2014*) because of the small sample size. We had an insufficient sample size to include covariates on occupancy, but we did include canopy cover and whether a survey occurred before, during, or after sunrise as covariates on detection probability. We used the canopy cover covariate as a proxy representing the different land use types we surveyed (agricultural, mixed, and forest), because canopy cover was generally higher at forested and non-agricultural sites. We also used detection probability as an indicator of abundance, by reasoning that a species was more detectable at an occupied site because multiple vocalizing individuals were present (*Royle & Nichols, 2003*).

Our model included fixed effects on species occupancy and hyperparameters on all detection probability parameters. Because we only surveyed focal species, we did not use data augmentation in the model. We fit a Bayesian model solved using a Markov Chain Monte Carlo (MCMC) algorithm (*Link et al., 2002*) implemented in JAGS (4.2.0, Plummer 2003) accessed via R statistical software (*R Core Team, 2016*) with the jagsUI package (*Kellner, 2015*). Uninformative priors were assumed for all parameters. Three independent chains of 10,000 samples were run with a burn-in period of 5,000 and a thinning rate of three. Effective mixing of these chains was assessed visually and by means of the Gelman–Rubin convergence statistic (<1.1; *Gelman et al., 2004*).

**Table 1  Habitat association and mean call number.** Mean call number for all three habitats including the standard error (SE), F-value, and P-value for all habitats. For all species, df total was 103, df factor was 2, and df error was 101.

| | Mean call number agriculture ± SE (# Calls) | Mean call number forest ± SE (# Calls) | Mean call number mixed ± SE (# Calls) | *F*-value | Agriculture-forest *P*-value | Agriculture-mixed *P*-value | Forest-mixed *P*-value |
|---|---|---|---|---|---|---|---|
| Red-vented Bulbul | 202.30 ± 28.98 | 141.78 ± 19.76 | 101.58 + 12.99 | 5.21 | 0.13 | 0.01 | 0.42 |
| Red Jungle Fowl | 34.14 ± 4.76 | 51.92 ± 6.45 | 29.33 ± 4.51 | 4.93 | 0.05 | 0.80 | 0.01 |
| Zebra Dove | 44.73 ± 7.81 | 4.58 ± 1.41 | 21.45 ± 5.07 | 13.79 | 0.00 | 0.01 | 0.09 |
| Common Myna | 20.57 ± 4.25 | 1.06 ± 0.46 | 3.00 ± 1.58 | 16.01 | 0.00 | 0.00 | 0.87 |
| Silvereye | 272.62 ± 37.07 | 269.97 ± 43.46 | 175.82 ± 27.63 | 2.13 | 1.00 | 0.16 | 0.18 |
| Common Waxbill | 41.05 ± 11.72 | 0.17 ± 0.17 | 0.52 ± 0.25 | 11.23 | 0.00 | 0.00 | 1.00 |
| Red-browed Firetail | 1.97 ± 0.68 | 4.58 ± 1.80 | 5.21 ± 1.87 | 1.30 | 0.44 | 0.30 | 0.96 |
| Mo'orean Kingfisher | 0.54 ± 0.19 | 0.58 ± 0.22 | 0.48 ± 0.32 | 0.04 | 0.99 | 0.99 | 0.96 |
| Grey-green Fruit Dove | 0.27 ± 0.11 | 0.44 ± 0.12 | 0.48 ± 0.15 | 0.85 | 0.58 | 0.45 | 0.97 |

## RESULTS

### Results for average calling number

The average number of calls per 5-minute survey was highest for two introduced species, silvereye and red-vented bulbul (Table 1). It was lowest for the two native species, Grey-green fruit dove and Mo'orean kingfisher. Common waxbill, zebra dove, and common myna all showed a significantly greater average call number per 5-minute survey in the agricultural sites ($p < 0.01$) whereas red jungle fowl was significantly more prominent in forested areas ($p < 0.05$) (Table 1). Red-vented bulbul, red jungle fowl, silvereye, red-browed firetail, Mo'orean kingfisher, and grey-green fruit dove showed no significant habitat association with the average number of calls data (Table 1). Zebra dove, common myna, and common waxbill showed a significant association with agricultural areas based on the number of total average calls at each site per 5 min (Table 1).

### Hierarchal clustering results

We found strong evidence of temporal partitioning in the vocalizations of native versus non-native species. In the assessment of 5-minute surveys, the hierarchical cluster analysis split detected species into two distinct groups that completely coincided with native versus non-native species (Fig. 3). For Mo'orean kingfisher, logistic regression confirmed its detection during a survey was negatively associated ($p = 0.016$) with the total number of non-native birds detected concurrently. On the other hand, there was no evidence of avoidance among the two native birds ($p = 0.444$)

### Occupancy modeling results

All but one of the species had occupancies >0.8, suggesting that these species were widespread throughout the study area (Fig. 4). Only grey-green fruit dove, a native species, had an estimate occupancy of <0.8, but it was still >0.6.

Different morning times explained differences in detection probability among non-native species (Fig. 5). All non-native species, except red-vented bulbul and red jungle

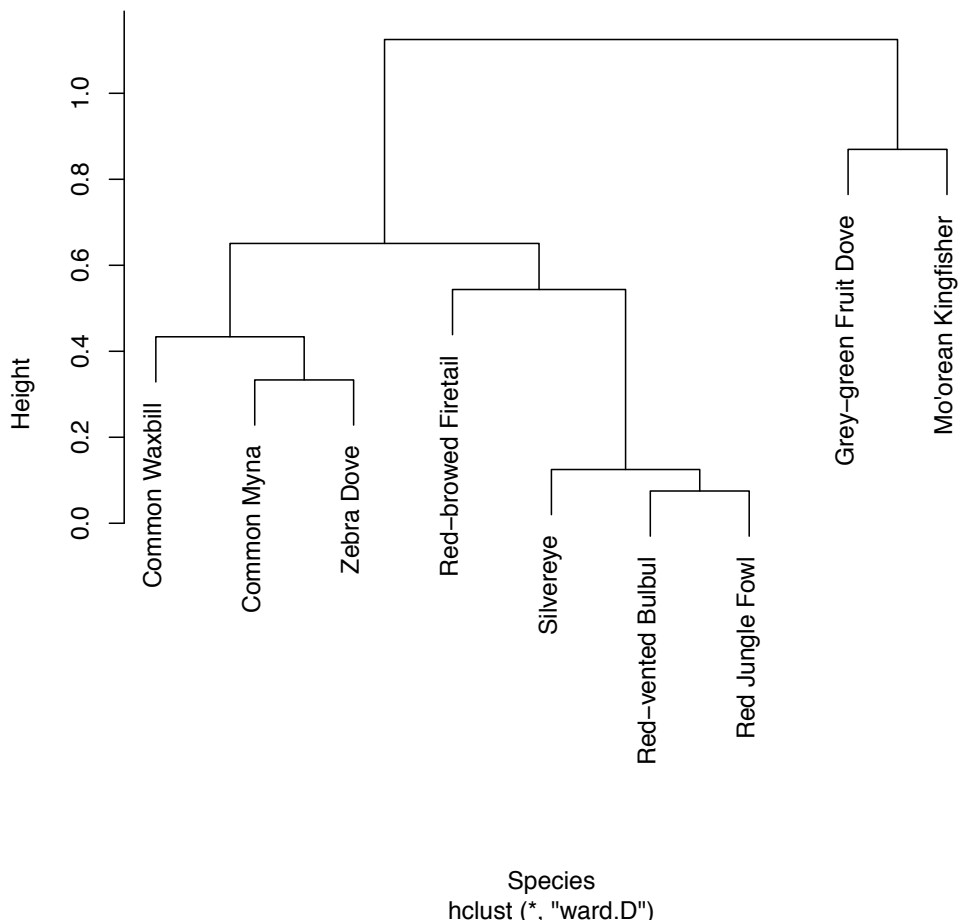

**Figure 3** **Hierarchical clustering of co-detection of nine bird species from five sites.** Hierarchical clustering was limited to the five sites at which all nine bird species were present. Temporal partitioning and vocal behavior of the nine birds are presented with this clustering.

fowl, displayed the same pattern whereby detection probability was higher after sunrise compared to before sunrise. In contrast, there was no discernable temporal pattern for the two native species; they did not call more or less at any of the morning times studied.

We identified a negative relationship between canopy cover and detection probability for three non-native species (common myna, common waxbill, and zebra dove), suggesting that these species may be more abundant in agricultural habitats (Fig. 6). Four non-native species (red-vented bulbul, red jungle fowl, silvereye, zebra dove) had the highest average detection probabilities per survey (>0.5), whereas both native species (grey-green fruit dove and Mo'orean kingfisher) were among the group with the lowest detection probabilities (<0.5) (Fig. 5). Taken together, the occupancy modeling findings suggest that, although widely distributed, native species may occur at lower levels of local abundance than non-native species.

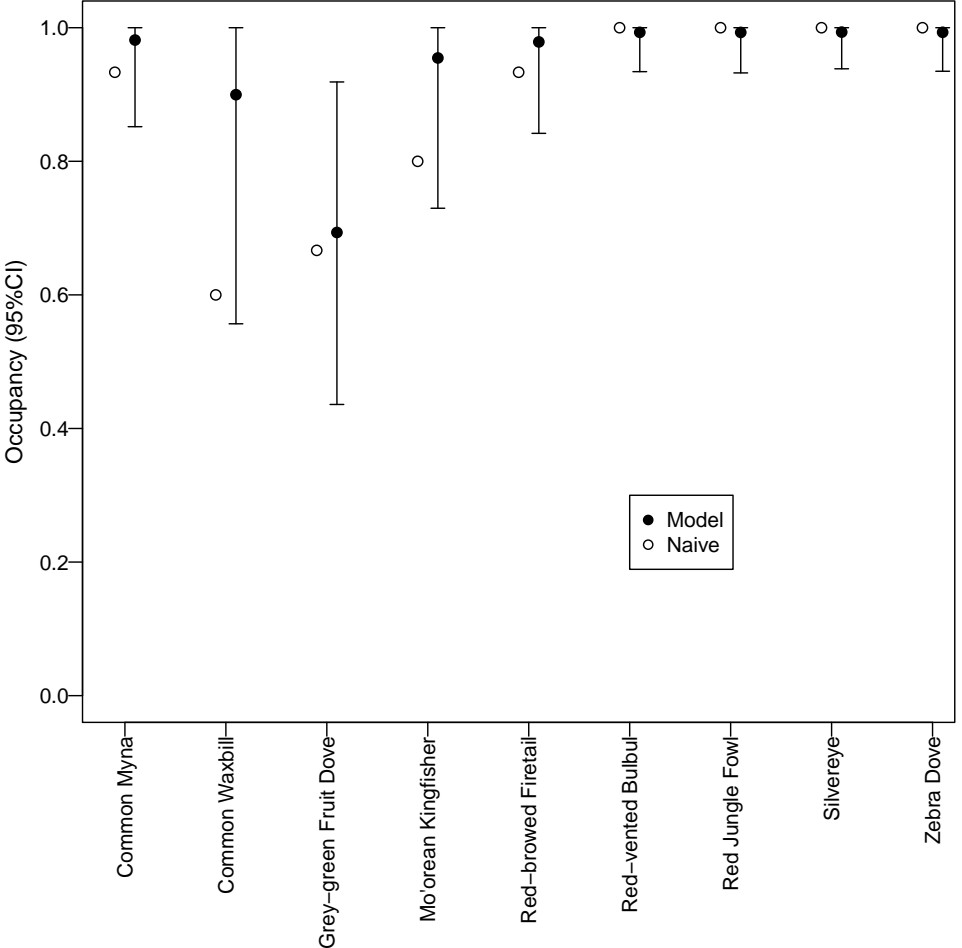

**Figure 4 Estimated occupancy.** Estimated occupancies of avian species surveyed using automated recorders. A multispecies occupancy model was used to address potential bias due to detection probabilities <1. Naive occupancy is the proportion of survey sites at which a species was detected.

## DISCUSSION

### Acoustic monitoring

Automated acoustic recording devices have been used in many different fields of biology to note acoustic activity of various species during different times of the year as well as different times of the day (*Jones et al., 2014*; *Baumann-Pickering et al., 2015*). These monitoring devices allowed us to efficiently repeat surveys at sites in remote areas that were difficult to access due to dense vegetation and steep inclines (*Frommolt, Bardeli & Clausen, 2008*). However, visual point counts in addition to acoustic recordings could be conducted in Mo'orea to further improve the accuracy and precision of our surveys (*Bibby et al., 2000*). Both sources of data could be included in the same occupancy model (*McGrann & Furnas, 2016*).

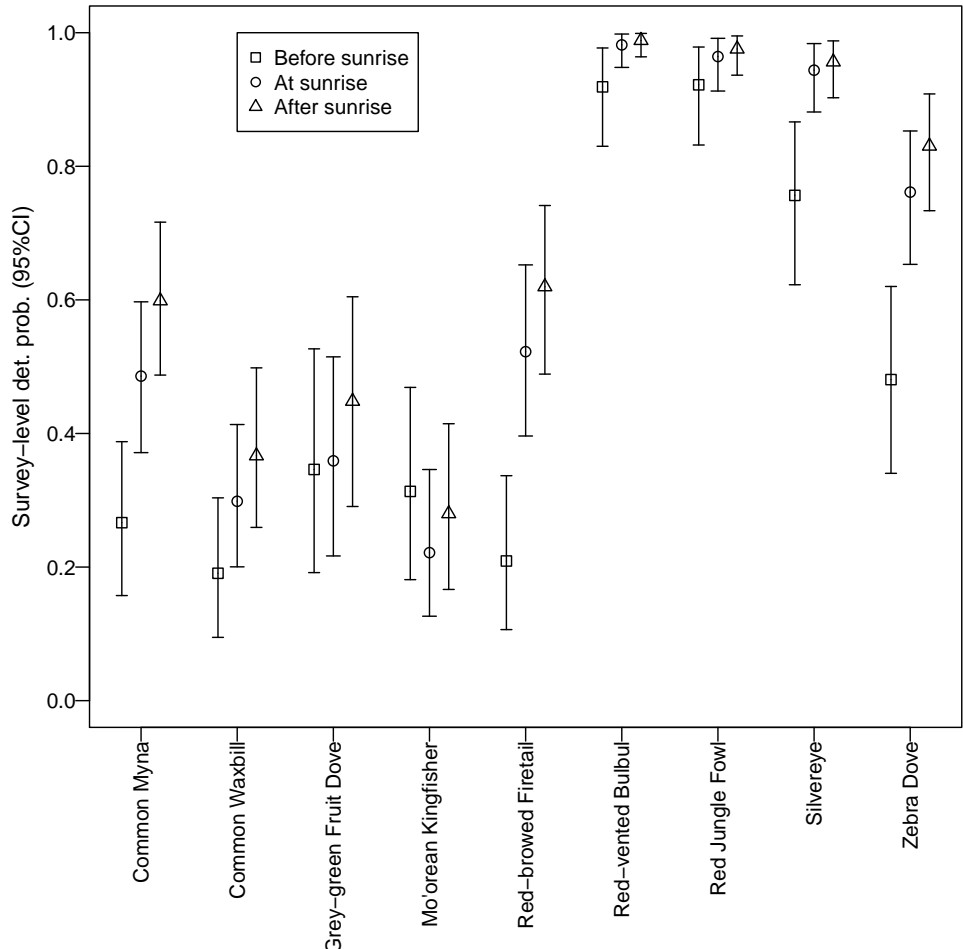

**Figure 5 Canopy cover detection probability.** Associations between forest canopy cover and detection probability for avian species surveyed using automated recorders. We deemed there to be an association when the credible interval for the parameter estimate did not overlap zero.

## Inference about activity and abundance

In this study, we found non-native birds calling at a significantly greater rate than native birds in each of the three habitats studied. Female and male avifauna in the tropics are known to call year-round; however, calling activity and call repertoire may change during breeding season (*Montgomerie & Weatherhead, 1988*; *Langmore, 1998*). The breeding seasons of the native and non-native avifauna on Mo'orea vary from species to species or are unknown entirely (*Spotswood, 2011*). Future studies should analyze if and when the calling activity of Mo'orean avifauna changes throughout breeding seasons.

Furthermore, our use of occupancy modeling helped us make inferences about occupancy and abundance for native and non-native species. Survey-level detection probability was lowest for the two native species, grey-green fruit dove and Mo'orean kingfisher. This difference suggests that, while all bird species were widely distributed across all three land-use types (agricultural, forested, and mixed), local abundance of native species may have been lower than non-native species at occupied sites. On the

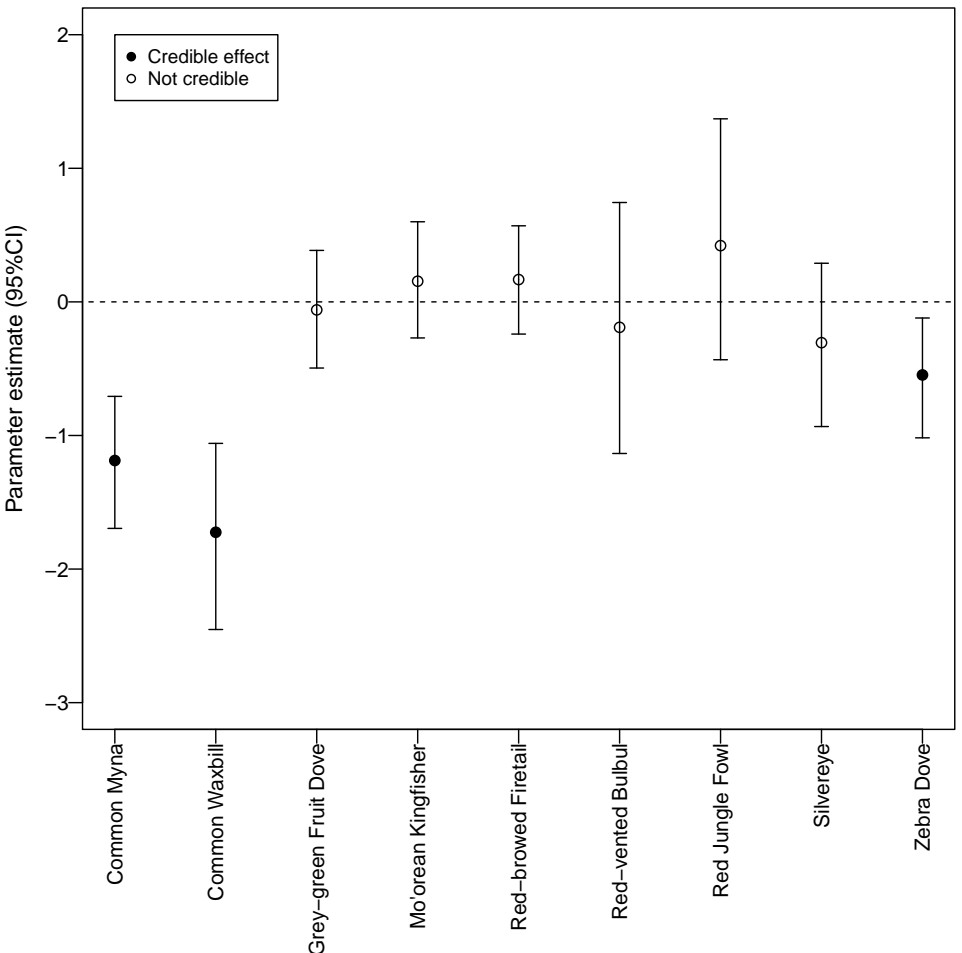

**Figure 6 Estimated detection probabilities.** Surveys were 5 min once a day repeated over 3 consecutive days commencing 30 min before sunrise, at sunrise, and 30 min after sunrise.

other hand, detection probability of three non-native species (zebra dove, common myna, and common waxbill) was significantly higher in low canopy habitats, suggesting higher abundances in agricultural areas.

One caveat that must be considered when linking detection probability to inferences about abundance (*Royle & Nichols, 2003*) is the sound transmission properties of different habitats. Lower detectability of three exotic species in higher canopy cover could be partially due to greater attenuation of calls in forested habitats (*Catchpole & Slater, 2008*). Low detectability of native birds due to sound attenuation, however, is unlikely. The call structure of native birds provides a greater likelihood of transmission because of the low frequency call of grey-green fruit dove and the high intensity, long duration call of Mo'orean kingfisher. Low frequency calls are more likely to travel through forest habitats (*Catchpole & Slater, 2008*), which may be a clue to the evolutionary and behavioral adaptations of the

native birds being adapted to live in a high-density forest area. Therefore, lower detection probabilities of native versus exotic birds may be evidence to reduced abundances for the former.

## Inferences about behavior

The results of hierarchical cluster analysis and logistic regression showed that Mo'orean kingfisher calling behavior was negatively associated with the presence of non-native calls. Mo'orean kingfisher calls lie in the same frequency band as all of the non-native species' calls. Mo'orean kingfisher calls could be obfuscated by the vast number of calls from the non-native birds (as high as 40 per minute for red-vented bulbul). Mo'orean kingfisher may not be calling as frequently because their songs are not effectively transmitted in the presence of the non-native calls, or because there is a lower density of Mo'orean kingfisher altogether. The obfuscation of the Mo'orean kingfisher call may adversely affect their social behavior, reproductive success, and overall population numbers.

Native bird calls may be less obfuscated by non-natives in the early morning. As seen in the morning time detectability results, native species showed no significant difference in calling times before and after sunrise, while non-native species were detected primarily after sunrise. Native passerines in Mo'orean perhaps avoid competition with non-natives by calling before sunrise when there is less non-native activity and less obfuscation of native calls. This result agrees with the notion of non-native species displacing native species from their behavioral niches (*Mooney & Hobbs, 2000*), in this case in terms of acoustic transmission.

Additional studies should be conducted to look into the morning calling times of native birds in areas where there are no non-native birds present, compared to areas with a loud non-native bird chorus to verify if natives are avoiding the non-native chorus. In addition, birds may be active in calling at dusk as well as dawn (*Zwart et al., 2014*). Calling patterns prior to dusk should be investigated to see if there is a different calling pattern for non-native and natives in the late afternoon.

## Ecology of non-native and native avifauna

The results of the present study demonstrate that non-native bird species thrive in all of the tested habitats consistent with prior studies showing non-natives prospering in disrupted habitats (*Mack et al., 2000*; *Gurevitch & Padilla, 2004*). The non-native common myna was significantly more active in the habitats disrupted by agricultural expansion. Interestingly, common myna removal projects have been initiated in the forests of Mo'orea's neighbor island, Tahiti, to protect native species (*BirdLife International, 2016*). Common myna removal projects may prove to be more effective if conducted in agricultural areas as well as forests, based on our findings that common mynas were less frequently detected in forests.

There is very little information about the habitat requirements, calling activity, and natural history of native avifauna in French Polynesia (*Coulombe, Kesler & Gouni, 2011*). Mo'orean kingfishers were shown in this study to have low call counts, low detection probability, and a negative correlation with non-native calling activity in all of the habitats studied. Future research should analyze habitat preferences of Mo'orean kingfishers in

greater detail. Closely related kingfisher species have been seen habituating coconut trees in managed and unmanaged coconut farms (*Coulombe, Kesler & Gouni, 2011*). Studies of the Mo'orean kingfisher should be conducted in coconut farms to support or refute this idea.

Grey-green fruit dove exhibited low calling activity, lower occupancy than other species, and low detection probabilities in all habitats studied. Grey-green fruit doves were more active in mixed habitats with moderate canopy cover; however, the differences were insignificant. Different canopy cover and habitats should be analyzed to see what the preferential habitat is for grey-green fruit dove so population restoration can be put into action.

## Conclusion

Multi-species occupancy modeling and automated acoustic recording devices are tools that could be used to increase the precision of parameter estimation for testing habitat and behavioral hypotheses. We applied them to provide the first comprehensive evaluation of avifauna on Mo'orea. Altogether our findings suggest that native species are avoiding calling when non-natives are active and that abundances of natives may be substantially lower than non-natives. Forests may provide a refuge for native species because non-natives were detected less frequently there. However, the conclusions of our study are limited by a small sample size. We recommend additional sampling using our methods as well as point counts. Additional surveys are necessary to create a clearer picture of the habitat preferences of Mo'orean kingfishers and grey-green fruit doves and the interactions of these species with non-native avifauna. This information will be essential for planning conservation actions protecting native birds in French Polynesia from endangerment or extinction.

## ACKNOWLEDGEMENTS

We would like to thank Dr. Justin Brashares, Dr. Cindy Looy, Dr. Patrick O'Grady, and Dr. Johnathan Stillman from University of California, Berkeley for their invaluable advice and support throughout this research study. Also, Scripps Institution of Oceanography's Dr. John Hildebrand and University of California, Berkeley's Dr. Justin Brashares for supplying us with automated acoustic recording devices. Without their generosity, this study would not have been possible. Thank you to Eric Armstrong and Ignacio Escalante (University of California, Berkeley) for their assistance with statistical analyses. We would like to acknowledge Natalie Stauffer-Olsen (University or California, Berkeley) for her guidance in writing of the manuscript. Thanks to the Gump Station for providing housing and company. Special thanks to Eric Lehmer for his support with making the maps in this paper. Finally, thank you to the fellow undergraduate student researchers that inspired us every day.

### Funding

The authors received no funding for this work.

## Competing Interests

The authors declare there are no competing interests.

## Author Contributions

- Vanessa M. ZoBell conceived and designed the experiments, performed the experiments, analyzed the data, contributed reagents/materials/analysis tools, wrote the paper, prepared figures and/or tables, reviewed drafts of the paper.
- Brett J. Furnas wrote the paper, prepared figures and/or tables, reviewed drafts of the paper.

## Animal Ethics

The following information was supplied relating to ethical approvals (i.e., approving body and any reference numbers):

This was an observational study and animals were not handled. This research was done with UC Berkeley Gump Station class. The protocol is uploaded as a Supplementary File.

## Data Availability

The raw data has been supplied as a Supplementary File.

## Supplemental Information

Supplemental information for this article can be found online at http://dx.doi.org/10.7717/peerj.3761#supplemental-information.

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
