# Peer review of "Impacts of land use and invasive species on native avifauna of Mo’orea, French Polynesia"

_PeerJ, doi:10.7717/peerj.3761_

## Round 0.1 · original submission · Major Revisions

Both reviewers have been very comprehensive in their criticisms of your paper and indicate a number of problems with the current manuscript. Reviewer 1, mostly indicated relatively minor issues that can be fixed up in a revised version taking the same approach as your current paper. However, Reviewer 2 has highlighted some concerns about the validity of your sample design and the conclusions that can be drawn from. These do appear to compromise the conclusions that you have reached currently, and I suggest that you consider changing the focus of your paper to perhaps reflect more on the findings outlined by the reviewer in the final section (comments to the author).

I encourage you to engage very thoroughly with the very constructive comments from both authors, even though this will require quite major revisions.

Reviewer 1 ·

Basic reporting

The paper uses a novel method to investigate the impact of different land cover types on the presence and abundance of native and introduced birds in Moorea, French Polynesia. It is important because it fills knowledge gaps in this remote island, gives us insight into the effects of introduced species, and presents an application of a method that will likely be heavily used in the near future.

I applaud the authors' application of the method on this reduced bird community, and acknowledge the importance of their findings for bird conservation in French Polynesia.

The science behind the article is sound, although some clarifications on the methods ans results are necessary (more below). However, the article would benefit greatly from another English proof-read from a native English speaker. Some of the sentences’ flow could be improved and some words and sentences could be replaced with adequate terms and concepts (e.g. “bird numbers” needs to be replaced with “abundance”). I suggest a through revision of the English before re-submitting.

Most of the article is written in third person. Nowadays, scientific publications are written in first person. I suggest you change your article to reflect this trend. “We studied…we found..”

As a reviewer, I would have appreciated a double-spaced document. It’s heavy on the eyes to read a single-spaced article like this one. Tip for next time.

Abstract
The abstract reads like a bullet point list that has been merged into a paragraph. Make sure to ensure flow of your sentences. Good summary but needs improvement of the English.

Introduction

The introduction works well for the general topic of forest vs agricultural areas and how this impacts birds. That is the broader problem the authors are dealing with and so it is valuable to have this introduction. I believe it would benefit from more information on island avifauna (examples from other oceanic islands and agriculture), and from a wider context of French Polynesia. What grows there? How long has agriculture been around on the island? Are there other studies about birds in the islands? The introduction to the specific study should come a little earlier in the introduction. The flow of some paragraphs could also be improved with some English proof-reading.

Line 16: improve sentence starting “The decline…” to something like “The homogenization of land cover caused by agriculture, has resulted in a decrease of habitat for many animals that rely on natural ecosystems”, or something along those lines.
Line 24-26: describe what type of agricultural expansion (type of crop, management techniques).
Line 29: conversion from “natural ecosystems” would be more appropriate as other ecosystems are also converted to cropland (like natural savannahs, wetlands)
Line 39: missing full stop.
Line 42: redundant sentence.
Line 50: sharp transition. Add a sentence that takes the reader from ecosystem services to monitoring techniques. Paragraph from lines 55-60 would be a good start.
Line 52: missing full stop.

Methods

Make sure all methods are written in first person and in past tense “we sampled…we found…we described…we used”. Many sentences are currently in the third person.

Line 71: fix Km2
Line 71: refer to figure of study area.
Line 74: do not capitalize Hibiscus if you’re not capitalizing other common names. Keep it standard for all whatever you decide to do.
Line 81: remove “the” before each species name, it is not necessary.
Line 87: why did you not sample the chestnut-breasted manikin? Write a sentence explaining this.
Line 90: what audio processing software did you use to make the spectrograms?
Line 97-98. Did you do a formal classification of the land cover types? From Google Earth you cannot process the imagery in ArcMap or other software, so which imagery did you use to construct land cover maps and then select your study sites?
Line 114 and 124: did you test the distance to which the recorder can record a sound? This is important to avoid counting an individual twice for two adjacent boxes.
Line 127: Figure 1 should have been cited in the first paragraph of study site.
Line 140: missing full stop.
Line 142: is a unique call considered a new individual? How do you determine that it is not the same individual calling?
Line 145: “coo”.
Line 152: you cite figure 6 after Figure 1…where are all the other figures referred to? You need to keep figure citing in order.
Line 156: missing parenthesis.
Line 165-185: I don’t think these descriptions are necessary if you are to include the spectrograms in the paper. I suggest removing this paragraph or putting it in supplemental materials.
Line 188: justify the use of a nested ANOVA
Line 194-197: improve sentence, as it is it confuses the reader.

Results

Line 215: I suggest starting the results section with a more general paragraph, then go into the specifics.
Lies 215-226: specify what the average calls are: is this per day? Or per sample?
Line 230: change “significance” for “significant”
Line 235: I think you mean Figure 12.
Line 240: I think you mean Figure 13. Check all other citations of these figures.
Line 243: “between canopy cover AND detection probability.

Discussion

The discussion hits some important points about the research and its implications for conservation, but it falls into too much detail for each species and it is disorganized. The subtitles help keep it in order, but even within the subtitles you could summarize some of the paragraphs and start the paragraphs by stating your main point and then explaining it in detail.

Line 254: replace “un-modelled” with original or raw.
Line 258: “frequency”.
Line 280: you need to support this statement. One way of doing this is to compare sites with presence of both native and invasive species, to sites with only native species (if any) to see if the native birds shift their singing time in response to invasive species presence.
Line 283: why are birds more prone to predation before sunrise?
Line 381: I have never known fossil records to be useful in knowing birds’ habitat preference, precisely because fossils are not common for birds. I don’t see the point of mentioning this, I would delete these two sentences from here as they add no useful information.
Line 396: I believe you mean “Least concern” as “Least threatened” is not an IUCN category.
Lines 393-417: I believe you’re trying to say that if we knew more about the Mo’orean Kingfisher and Gray-green Dove, they might be classified in a higher threat category. I agree with this statement and I think it should be made clear…as it is it’s confusing. Re-write these sentences to reflect this proposal. On a different note: make sure you alert IUCN though the Endangered Bird Forums about the scarcity of these two birds in your study. This is valuable information from an understudied place that I’m sure will be much appreciated.
Line 443: the country’s name is ColOmbia, not Columbia.

References
Please check that all your references are in the text also. I didn't check each one but have the feeling that some of them were not cited in the text. This needs to be in order before publishing the paper.

Line 472: incomplete reference.
Line 529: some references are indented and some are not. Check for consistency.
Line 600: why do you have 3 IUCN citations? You only need one and I believe you only cited one.

Figures
Figure 1: this figure needs a lot of work to be better for the reader. First thing is to create a reference inset in which you can direct the reader from the general landscape to the specific valley. Having the landcover or a background satellite image would really help the reader see the landscape context. It would also be useful to have a close-up map of just the sampling sites overlaid on a land cover map to see the representation of different lad cover types in the study sites. Decrease size of N arrow, add color legend on figure instead of in caption.
Here’s an example from a PeerJ paper:

Figure 6-10: I don’t think it’s necessary to have each call be its own figure. I suggest one of two things: make a multi-panel figure with all the calls, or leave one figure as example in the paper and add the rest as supplementary material.
Figure 12: I recommend color-coding invasive and native species so the differences in detection are more obvious to the reader’s eye.

Experimental design

The experimental design is sound and fit for the question being asked. The research fills a knowledge gap and uses a novel method successfully. Anyone could replicate the study if needed.

The one comment I have about the experimental design is that the sampling period was too short. A mere 3 mornings per site (45 mins) seems too short to make decisive conclusions about the avifauna of the whole island. I would have liked to see a sampling design that took into account seasonality, recording birds at different times of year, and several times.I understand that this would mean longer analysis time, but it would make the conclusions stronger and more applicable.

Validity of the findings

No comment

Additional comments

The research is very interesting and would be well received by the ornithological and conservation audiences. I have made some comments regarding how you present and discuss your results. In general, the ideas are a little disorganized in long, confusing paragraphs that could be summarized. I mostly suggest a strong re-write and revision of the English throughout to improve the readability of the paper and better highlight the important results you present.

·

Basic reporting

Text would benefit from considerable editing. For example, the first three sentences of Results (lines 215 to 217).
“Activity in calls from the forest was most frequent for the invasive silvereye with approximately an average of 750 calls in the forest and the invasive red-vented bulbul with over 550 total average calls in the agriculture areas (Fig 11). The least frequent activity was found in the two native birds. Grey-green fruit dove had a maximum number of average calls of 1.5 in the mixed sites.”
May be better presented as -
“Cues were most commonly recorded for introduced species, Silvereye and Red-vented Bulbul (c750 and 550 cues in forested and agricultural areas respectively). Cues for the native species, Grey-green Fruit Dove and Moorean Kingfisher were recorded on 1.5 or less occasions in mixed, and forested habitats respectively (see Figure 11).”

The text to go with Figures is overly long, not clear, and replicated from Figure 2 to 10 with only the name of the species involved being changed between figures. Need to resolve that - or relegate these figures to an Annex.

In the introduction section there is an over emphasis on the negative impact of agriculture. The introduction could be significantly reduced if it was restricted to issues of relevance to the agricultural system on Moorea.

There needs to be a greater consideration in the Introduction of the site – it is a priority site for birds http://datazone.birdlife.org/site/factsheet/23759, for which 4 native species are identified as trigger species. Two of these trigger species are not discussed in this paper – the swiftlet is not well surveyed by acoustic recording, while the Tahiti Petrel is best surveyed nocturnally to the times reported in the survey – and are not likely to be in the valleys.

There needs to be more discussion regarding the choice of method used - and justification of this given that it is a non-standard method. Lines 55-60 overplay the use of acoustic recordings and doesn't consider the negative consequences of using only acoustic recordings. .

Experimental design

• It is not explained how the method is an improvement over point counts, for which there is a half century of expertise. In particular, Klingbeil and Willig (2015), note that “if short-term monitoring is the goal point counts are likely to perform better than ARUs (automated recording units), especially if species are rare, or vocalise infrequently”.
• In comparison with other studies that have used acoustic recorders, studies elsewhere have been either
o for a longer period of time, primarily to monitor trends in numbers,
o at a much higher rate than indicated in this paper (eg Furnas and Callas 2014, Kagu surveys in New Caledonia, the hunt for Tooth-billed Pigeon in Samoa) or
o used in an array system to estimate bird densities by summing the location of different calling birds (eg Dawson, D K., and Efford.M E "Bird population density estimated from acoustic signals." Journal of Applied Ecology 46.6 (2009): 1201-1209)
• The method used here appears not to utilise any of the strengths of the recording system – with the exception that it provides a permanent record of what was calling, which can be validated by a future surveyor. I presume the files are stored for posterity.
• The analysis was undertaken by manual assessment rather than any form of automated identification There is no discussion about the weaknesses of the method (such as sightings of birds that don’t call frequently (eg swiftlets) or birds that call at different times of day (eg Petrels).
• The use of cues would have been better explained if the cue used for each species had been presented in the Figures. Do all the Figures represent 1 cue for each species –are the individual notes counted or is just one of the notes used as the ‘cue’?
There is no attempt to explain whether different recording instruments had a bearing on the results. Was there variation in the analyses of the outputs from the different types of recorder? Or was any impact avoided by randomly choosingthe location for the acoustic recorder c/w the

Validity of the findings

1) The use of detectability as a comparison between species is inappropriate. The nature of the cue that the author uses to count the presence of a species, varies significantly between species. Even the use of detectability WITHIN species but between habitats is also questionable – although the author does acknowledge this.
a) IF detectability was to be used then the next step would have been to undertake an assessment of the number of cues per unit time per individual, for the different species. For Ptilinopus species, for instance, an individual is likely to call, and so activate a cue, on no more than 1 or 2 occasions in a 5 minute period – while a Silvereye might utter several 10s of cues while in range – but, conversely, will move in and out of range within a 5 minute period.
b) There is no discussion about the range of the acoustic equipment beyond a reference to another paper in a completely different habitat, with a completely different variety and density of species. The detectability of a species as measured here will be different for a quiet/soft species c/w a raucous species. In fairness the 2 native species are unlikely to be in the quiet/soft category. The author notes in the spectrogram that ‘the warmer the color corresponds to a more intense sound’ (line 160). However, there is no discussion about whether there is a cutoff when sounds are not logged – presumably when the colour change doesnt show - and how that relates to distance. One problem with recordings as overload caused by multiple songsters of multiple species at the same time - it then becomes difficult to disentangle - particularly if spectrograms are the main means of assessment..
2) There is a lot of discussion about agriculture, impact of invasive species, etc, etc in the discussion - which is not justified by the results or findings.

Additional comments

The most useful information presented here is the variation in cues between habitats for a given species (notwithstanding the issues of differences in detectability between habitats).
a) The fact that, as the author notes, 3 introduced species are associated primarily with non-native, agricultural, habitats – with very little occurrence in forest or mixed habitats - is useful supporting evidence to other studies, and is unlikely to have been affected to this degree simply by differences in acoustic variations between habitat. Given the cost and effort being invested in removing mynas and bulbuls from forested habitats in the neighbouring Tahiti this is worth highlighting.
b) My interpretation of the Kingfisher graph in Fig 11 is that there is no evidence that the species shows preference for native habitats (the Fruit-dove shows a non-significant avoidance of Agriculture) – accordingly this paper provides no evidence that the kingfisher is likely to be impacted by habitat modification, while there is some evidence that Fruit-doves avoid forested areas. Both of these conclusions are in line with the distribution of Todiramphus and Ptilinopus species elsewhere in the Pacific, yet contrasts with the BirdLife definition that suggests the Kingfisher is Highly dependent http://datazone.birdlife.org/species/factsheet/moorea-kingfisher-todiramphus-youngi/details , but the Fruit-dove only Medium Dependent on forest habitats.

---

## Round 0.2 · Minor Revisions

As a result of the major revisions that you undertook, I felt that the manuscript needed to go back to reviewers. Unfortunately neither of the original reviewers were available to take a look but I have managed to solicit a review from a new reviewer. The comments from that reviewer are generally very positive and just require additional changes to the text. I believe that these changes are necessary in helping to describe your methodology and describing your findings clearly.

·

Basic reporting

This needs improving.

Introduction: This section has reasonably good content, however, I feel like it could benefit from a little more organization. I suggest the authors consider the “story” that they are trying to tell, and organize the paragraph order and information in the paragraphs around it.I think the introduction should make it clear what the hypotheses and predictions are in regards to avian abundance and environmental change/ land use as the authors state that importance of this in both the abstract and earlier in the introduction.
Methods: This section was generally written clearly. However, there are a few details missing that make it difficult to replicate the study.
Results: reporting needs to be clearer. I little bit disjunct a times.
Discussion:this section needs to be better organized for flow and tightened up. However, lots of good concepts and discussion point. I also feel like there needs to be more discussion connecting this study to past studies on a) using acoustic units as a proxy for bird presence/ abundance and 2) conservation/ ecological issues of the two native species, and again tie this to previous tropical studies where native doves (e.g. Doves in Madagascar, Grenada) or kingfishers or (other freshwater piscivorous birds) have been studied.

Experimental design

There needs to be more details on how some field measurements were taken. There also needs to be more details/ discussion on the use of ARUs as a proxy for bird presence as some birds may be seen and not heard (and vice versa), especially because this survey was a first avian survey in the area and therefore no one knows how point counts/ transect ground truth methods compare to acoustic detections.

Validity of the findings

Data is reasonably robust, although I would like to see more reporting of statistics information (not just p-values) stat. However, I also think that the authors really need to make it clear that they are using acoustic recordings to record the presence of a bird species and how the occupancy models deals with call counts. Also there needs to be more discussion on the limitations of the acoustic data so that the reader can temper their interpretation of the results (i.e. absence of calls does not necessarily equal absence of bird)

Additional comments

Please see the accompanying document for general and specific comments.

---

## Round 0.3 · accepted · Accept

Thank you for dealing with all of the comments raised by the final reviewer. The manuscript is much clearer as a result of these changes and is now acceptable for publication.